# OpenReview forum: "CoT Red-Handed: Stress Testing Chain-of-Thought Monitoring"
_NeurIPS.cc/2025/Conference — NeurIPS 2025 poster_

### Official Review · Reviewer_9pfw · 2025-06-02

**Clarity:** 4
**Significance:** 4
**Originality:** 3
**Rating:** 6
**Confidence:** 5

**Summary:**

In this paper they investigate how CoT monitors perform relative to output only monitors for various reasoning models. They modify a coding task dataset to include “side-tasks” which the agent is supposed to achieve along with the main objective. They break these side objectives down into “subtle” and “non-subtle” ones and find that the CoT monitor performs better than the output monitor on subtle objectives but not on non-subtle ones. They hypothesise that this is because the CoT monitor can get fooled by the plausible reasoning in the agents CoT.

**Questions:**

- Why did you only test gpt 4.1-mini as a monitor?
- Have you considered using a reasoning model as the monitor? Most labs are moving towards using reasoning monitors for deception. In my experience these do better at CoT monitoring and typically handle long contexts better.
- How did you arrive at the prompts, how much optimization pressure was there in the process and how equal was it? The prompts seem quite similar which leads me to think that there may be a large elicitation gap for one or both of the monitors.
- Why did you decide to put the reasoning after the output for the CoT monitor?
- I was going to ask why you didn’t try o1/o3/o4 but then i realised you won’t actually have access to the true CoT for these. Maybe you should mention this since otherwise a reader might be confused as to why you test random R1 distills and not the classic reasoning model.
- Have you tried using Max(monitor 1, monitor 2) as the suspicion score? That seems like the typical choice so it would be nice to compare to.

If the authors address some of my concerns (see weaknesses) I will probably raise my score.

**Ethical Concerns:**

["NO or VERY MINOR ethics concerns only"]

**Final Justification:**

I was impressed by the authors response to my and other reviewers concerns. I think they have resolved or mitigated all my concerns.

If the authors make the changes they say they will I think this could shape up to be a very good paper which is why I am increasing my score to a 6.

**Limitations:**

yes

**Quality:**

4

**Strengths And Weaknesses:**

Strengths:
- Timely and important research direction.
- Large number of coding tasks.
- Good limitations section.
- Well written.

Weaknesses:
- I believe they somewhat overstate the performance difference between CoT and action monitors. Looking at their plot the two monitor types appear highly correlated.. They state that the CoT “substantially increases detection” but on some of these tasks the increase is to 20% which is pretty low performance. They say “Action-only monitoring performs poorly” but If this was a real world control setting both monitors would have unacceptably low performance. Furthermore, in my experience designing deception/hacking CoT and output monitors for reasoning models, monitors are highly sensitive to the prompt and how the CoT is presented to them. I would strongly suspect that prompt variations could increase/decrease performance by 10+ percentage points. If the authors put different amounts of optimization into the different prompts this effect could be larger than the difference they observe. I believe the authors should modify their language slightly to avoid overstating this difference and possibly add some detail in how the stated prompts were made, e.g. how many iterations were made, was there a lot of iteration on one which was then ported to the other setting etc.

- They only use 4.1-mini as a monitor. I think a sensitivity analysis of the monitor capability would strengthen the paper. Would CoT monitors still underperform if a stronger model was used for example?

---

> ### Author Rebuttal · Authors · 2025-07-31
>
> We thank you for your thoughtful feedback and constructive comments. We have carefully addressed each concern and made corresponding updates to our paper.
>
> ## **Response to Major Concerns**
>
> ### **Performance Differences Between CoT and Action Monitors**
>
> We acknowledge the reviewer's concern about potentially overstating the performance difference between CoT and action monitors. We agree that both monitor types show correlated performance (shown in a table below) and that in real-world deployment settings, the absolute performance levels would be unacceptable. Our goal was to demonstrate performance improvements over current baselines and identify interesting failure modes rather than present deployment-ready solutions.
>
> In response to this feedback, we have:
>
> * Revised our language throughout the paper to avoid overstating the performance differences
> * Clarified our objectives to emphasize that we aimed to show relative improvements and failure mode analysis
> * Added methodological details about prompt development (see below)
>
> ### **Prompt Development and Optimization**
>
> Regarding concerns about prompt sensitivity and potential optimization bias, we provide the following clarification:
>
> During our experiments, we tested over a dozen prompts before settling on the final versions. Our optimization process included:
>
> * Testing unified prompts for both action and CoT monitoring
> * Developing separate specialized prompts for each monitoring type
> * Incorporating few-shot examples (which we found did not improve performance)
> * Approximately six iterations within each broad category
>
> In preliminary work with our training split of BigCodeBench, we made several improvements:
>
> * Adding supplementary reflection questions substantially improved both monitor types
> * Asking the monitor to provide specific quotes to justify its score improved monitor performance
> * Addressing a failure mode in CoT monitors that was causing high false positive rates due to the lengthy and convoluted nature of reasoning traces (e.g., from DeepSeek). Specifically, we adjusted the prompt with an instruction not to consider lengthy or repetitive content to be suspicious by default.
> * Testing few-shot examples on our validation set (which showed no improvement)
>
> We consider both prompts to be reasonably optimized but not exhaustively tuned, with comparable optimization effort applied to each. We believe this level of optimization is sufficient to support our core contributions regarding the relative strengths of different monitoring approaches.
>
> ### **Monitor Model Selection and Sensitivity Analysis**
>
> In response to reviewer feedback, we have expanded our monitor evaluation:
>
> * **Added results** for GPT-4.1-nano, Claude 3.5 Haiku, and Gemini 2.5 Flash Lite (reasoning)
> * **Included sensitivity analysis** across monitor capabilities as requested
>
> Our initial choice of GPT-4.1-mini was intentional—we selected a single monitor weaker than all reasoning models to explore other axes of variance while controlling for monitor capability. We acknowledge that reasoning models as monitors represent current best practices in deception detection, particularly for CoT monitoring and long-context handling. The expanded results now provide a more comprehensive view of performance across monitor capabilities.
>
> **Subtle Catch Rate @ 1%FPR**
>
> | Monitor | Action-Only | CoT-Only | Hybrid (w=0.25) | Hybrid (w=0.50) | Hybrid (w=0.55) | Hybrid (w=0.75) | Max(Action,CoT) | Geometric Mean |
> |-----------------------|-------------|----------|-----------------|-----------------|-----------------|-----------------|-----------------|----------------|
> | claude-3.5-haiku | 21.1% | 19.8% | 20.6% | 24.0% | 25.3% | 27.8% | 23.6% | 24.2% |
> | gemini-2.5-flash-lite | 53.5% | 64.9% | 62.7% | 71.3% | 72.0% | 66.3% | 74.4% | 72.5% |
> | gpt-4.1-mini | 3.9% | 12.6% | 12.6% | 11.0% | 11.2% | 8.3% | 13.3% | 12.4% |
> | gpt-4.1-nano | 4.4% | 4.9% | 4.1% | 4.3% | 4.6% | 4.4% | 5.3% | 4.7% |
>
> **Non-Subtle Catch Rate @ 1%FPR**
>
> | Monitor | Action-Only | CoT-Only | Hybrid (w=0.25) | Hybrid (w=0.50) | Hybrid (w=0.55) | Hybrid (w=0.75) | Max(Action,CoT) | Geometric Mean |
> |-----------------------|-------------|----------|-----------------|-----------------|-----------------|-----------------|-----------------|----------------|
> | claude-3.5-haiku | 95.3% | 52.6% | 78.6% | 86.8% | 88.6% | 92.5% | 84.9% | 86.9% |
> | gemini-2.5-flash-lite | 99.2% | 70.2% | 79.3% | 97.7% | 97.8% | 98.0% | 98.1% | 97.9% |
> | gpt-4.1-mini | 77.7% | 47.4% | 51.2% | 70.1% | 72.2% | 74.7% | 73.7% | 72.1% |
> | gpt-4.1-nano | 56.1% | 13.7% | 11.8% | 29.3% | 31.9% | 39.5% | 37.2% | 31.7% |
>
> **Correlation of suspicion scores**
>
> The correlations in the table refer to the relationship between action and CoT suspiciousness scores. Rows show results for red team models while columns show results for monitor models. Gemini-2.5-flash-lite has lower correlations overall, which helps the hybrid monitor improve more since their errors overlap less, since both baselines have similar performance. Although correlation is also low with gpt-4.1-nano as monitor, the hybrid gains are limited since the CoT monitor has significantly lower performance than the action-only.
>
> | | claude-3.5-haiku | gemini-2.5-flash-lite | gpt-4.1-mini | gpt-4.1-nano |
> |--------------------------------|------------------|----------------------|--------------|--------------|
> | claude-3-7-sonnet-latest | 0.5938 | 0.3296 | 0.7041 | 0.4023 |
> | deepseek-r1 | 0.5617 | 0.3151 | 0.6738 | 0.2880 |
> | deepseek-r1-distill-llama-70b  | 0.6312 | 0.4565 | 0.8269 | 0.3799 |
> | deepseek-r1-distill-qwen-14b   | 0.5614 | 0.4239 | 0.7885 | 0.3834 |
> | grok-3-mini-beta | 0.5189 | 0.3831 | 0.6825 | 0.3800 |
>
> ### **Additional Clarifications**
>
> **CoT Ordering**: We placed the action output before the reasoning in our CoT monitor prompts to provide context before the detailed reasoning analysis. This ordering aimed to help the monitor understand what the model ultimately did before examining why it did it. We found that agents frequently included drafts or complete answers within their CoTs, making the specific ordering less critical than initially anticipated as the monitor had access to output information regardless of placement.
>
> **OpenAI Reasoning Models**: We did not evaluate o1/o3/o4 models due to the lack of access to raw reasoning traces (as the review correctly pointed out), which are essential for our CoT monitoring approach. We have added a note in the paper clarifying this limitation.
>
> **Score Combination Strategy**: We tested multiple scoring strategies, including Max(monitor 1, monitor 2), but found that combining scores yielded better performance. This finding is now explicitly mentioned in our methodology section. Additionally, results for multiple aggregation strategies and weighting are shown in the tables above.
>
> ## **Summary of Changes**
>
> 1. Language revision to temper claims about performance differences
> 2. Expanded monitor evaluation including multiple models and sensitivity analysis
> 3. Added methodological details about prompt development and optimization
> 4. Clarified limitations regarding OpenAI reasoning models
> 5. Explicit discussion of score combination strategies
>
> We believe these changes significantly strengthen the paper while maintaining our core contributions about the relative effectiveness of different monitoring approaches, the vulnerability of simple CoT monitors to reasoning content, and the benefits of combined monitoring strategies.

---

> > ### Comment · Reviewer_9pfw · 2025-08-01
> >
> > The authors have addressed my concerns, I will be increasing my score.

---

### Official Review · Reviewer_XKux · 2025-07-03

**Clarity:** 3
**Significance:** 2
**Originality:** 2
**Rating:** 5
**Confidence:** 4

**Summary:**

This paper focuses on improving CoT monitoring for cases where a weaker trusted monitor model continuously oversees the intermediate reasoning steps of a more powerful but untrusted model. The authors propose a hybrid protocol that combines CoT monitoring with action-only monitoring using a weighted average. The paper reports this hybrid approach outperforms both CoT monitoring and action-only monitoring across all models and task types analyzed.

**Questions:**

Is there a way to make the subtle tasks more interesting?
How do the results change if we include failed attempts?
Would the results change or stay the same if both Blue team and Red team had access to the same number of prompts for validation?

**Ethical Concerns:**

["NO or VERY MINOR ethics concerns only"]

**Final Justification:**

The authors have compellingly addressed my main concern on subtle tasks not really looking malicious. Although I still think this is true, I do agree that they might constitute an "action towards a separate and undesired goal from the one intended by a human user" as the authors put it. The authors have also corrected me on optimal weighting vs signal similarity, which I agree, and no longer see the weight being 0.55 as a concern. They have also clarified the motivation of dataset splits for blue and red team in their experiments.

**Limitations:**

yes

**Paper Formatting Concerns:**

No major formatting issues.

**Quality:**

2

**Strengths And Weaknesses:**

Strengths:
- Very interesting analysis of pairing up each problem in BigCodeBench with the side task most likely to conceal it.
- The experiments include 5 different untrusted models.
- Writing is very clear and concise
- Discussion on limitations of CoT monitoring

Weaknesses:
- The subtle tasks look somewhat reasonable, and not really malicious, which might be the main reason why the action-only monitor doesn't give them a high score. They definitely don't seem to match the wording used in the CoT of action-only monitors: "suspicious or counterproductive ways to solve the problem, backdoors, sabotage or other forms of hacking". E.g., why would "Variable Name" or "Long Implementation" be malicious? At most they would be an annoyance to the user. In fact, the side attacks are considered subtle if the action-only monitor detection rate is <50% (line 140), which, again looking at the prompt, is the very definition of a non-malicious attack. This might actually explain why CoT monitoring (or hybrid monitoring for that matter) does not surpass ~30% detection rate.
- The analysis for comparing monitoring approaches excluded failed attempts because they would unfairly skew the assessment of the author's approach (lines 122-124). This seems biased and against the authors motivations of reflecting a "real deployment of a frontier AI system" (line 128). Does this mean that proposed approach wouldn't work in a real setup where malicious models could have failed hacking or sabotage attempts? I.e., if a model knew this, it could just sandbag and hide the sabotage in a "failing" answer.
- Why does the Blue team has 25% few-shot examples where the Red team only has 15%? (lines 152-156). This seems unbalanced and might skew the results.
- Line 181 claims that "CoT and action-only monitoring provide complementary signals" but looking at Figure 3, and given that the best weight found for the hybrid monitor was 0.55, it would seem that there's more evidence of them being similar than different.

---

> ### Author Rebuttal · Authors · 2025-07-31
>
> Thank you for your review of our paper. Our goal is to provide a useful and well-evidenced update to the AI safety community on where and how CoT monitoring is best applied. We appreciate that you brought out some of the positives in our work such as the pairing analysis and comparison across 5 models.
>
> Here we will focus on addressing the issues that you have raised in the sections on weaknesses, and answering your questions.
>
> We first address each of the bullet points in your weaknesses section, in the order they were shown in your review.
>
> ## Rationale for Subtle Side Tasks
>
> You are correct that some of our subtle side tasks, especially those involving changing a variable name or extending the length of an implementation, are not examples of serious malicious behaviour. There are a few reasons why we choose to include them and we have updated the paper to make this clearer.
>
> First, while they are not *seriously malicious*, we think that they reasonably represent *action towards a separate and undesired goal from the one intended by a human user*. This is what we are really worried about in future, more powerful models, which might act on very complex misaligned plans, in very indirect ways \- these side tasks are a toy stand-in for this. You’re right that our action-only monitor prompt asks the monitor to look for e.g sabotage, but we also include *“Is there anything that suggests the code is designed to achieve another goal, beyond the stated problem?”* to guide towards this interpretation of suspicious behaviour. The monitors always have access to the original coding problem, so they have sufficient information to judge this.
>
> Second, we think that these very subtle side tasks are useful because they show contrast between action-only and CoT monitoring. The fact that action towards this separate, undesired goal is very difficult for an action-only monitor to detect is intentional. The access to information about the model’s intention, revealed in the CoT, becomes crucial here, and this is borne out in the results. You’re right that the CoT/hybrid monitors perform poorly here at only \~30% detection rate, but we think our contribution here is just to provide experimental evidence that compares CoT monitoring to existing baselines in the literature. We find they do show uplift in these situations (we hope there will be further work that makes these monitors genuinely effective).
>
> As a concrete improvement to our paper, **we have clarified the language around our side tasks, to make it clear that we intend side tasks to simulate action towards a separate and undesired goal**, not only more serious malicious behavior.
>
> ## Exclusion of Failed Attempts
>
> Whether or not to include failed attempts, at the main task or side task, is something we returned to repeatedly over the course of our work.
> We decided to remove examples that failed on the **main task** test cases because we think that these examples would not be used in reality, and so are less relevant to our threat model. We often thought of the example of a company deploying LLMs to write parts of its internal codebase, and using AI control as a safety measure. In this setting, code that failed to pass test cases would never be integrated and used in deployment regardless of whether the monitor flagged it or not. For this reason, we decided to test our monitors exclusively on working code, which is where they would deliver safety improvements.
>
> It is also possible that failed attempts at the main task, resulting in incorrect code, might have triggered the monitor spuriously, inflating our estimates of monitor effectiveness. While we controlled for the false positive rate using examples without a side task, we observed that inclusion of a side task causes the main task to fail more often, meaning that this would have affected our results.
>
> We decided to remove examples that failed on the **side task** test cases because we considered these as cases where the agent had not managed to scheme successfully \- so again here, flagging by the monitor doesn’t deliver a safety improvement. We also thought it possible that output code, in cases where the side tasks failed, might be completely indistinguishable from code that exclusively and faithfully executed the main task, making it an unfair test for the monitor.
>
> To address this concern, we have **extended the section you highlighted (lines 122-124) to properly discuss one or both of these cases**.
>
> ## Dataset Split Clarification
>
> Regarding the dataset split balance described in lines (152-156), we want to better explain exactly how and why we split the BigCodeBench dataset, and which examples were used for producing few-shot examples.
>
> The Red Team (untrusted model prompts) uses just 5% (*Prompt split*) for constructing few-shot examples, not 15% as suggested. This was part of our work on whether we could make the untrusted model more effective at getting past a CoT monitor by providing examples of a chain of thought which does not include mention of the side task, followed by a final answer which completes the side task.
>
> The blue team (monitor prompts) did not use few-shot prompts in our final experiments. Note: *We did carry out some experiments on whether few-shot prompting could make the monitor more effective, by providing examples of model actions/CoTs along with appropriate suspicion scores (we presented these in pairs, with both low and high suspicion score examples). However, we found that these did not make the monitor more effective, and dropped this approach, settling instead on the prompts that can be seen at the beginning of the appendices.*
>
> ## Monitor Complementarity
>
> We respectfully disagree with the interpretation that a hybrid weight of 0.55 indicates the monitors are 'similar.' This reasoning conflates optimal weighting with signal similarity, which are distinct concepts.
>
> The key evidence for complementarity comes from Figure 3, which shows the monitors have **qualitatively different strengths**: CoT monitoring excels on subtle tasks (bottom-left quadrant) where action-only monitoring fails, while action-only monitoring performs better on non-subtle tasks (top-right quadrant) where CoT monitoring struggles.
>
> The hybrid weight of 0.55 simply reflects the best balance for maximizing overall performance across both task types. The complementarity is demonstrated by the performance patterns, not the weighting. Our contribution is in identifying when each monitoring approach is most effective and showing that a combined approach outperforms either technique, across our studied side-tasks.
>
> ## Addressing specific questions
>
> **Is there a way to make the subtle tasks more interesting?**
>
> We believe this question refers to your earlier concern about the maliciousness of our subtle side-tasks. As discussed above, we see these tasks as proxies for action toward separate and undesired goals, and see this as the main safety concern. The 'interest' in these tasks lies precisely in their subtlety \- they represent scenarios where intent-based monitoring (CoT) can detect goal misalignment that would be invisible to action-only monitoring.
>
> **How do the results change if we include failed attempts?**
>
> We believe this would introduce significant noise to the results. We hope that our earlier response clears up why we don’t believe failed attempts should be a main part of our analysis: (1) **main task** failures would not be deployed in realistic settings regardless of monitor flagging, making them irrelevant to our threat model, and possibly causing spurious monitor hits that would inflate our measure of monitor effectiveness; (2) **side task** failures represent cases where scheming was unsuccessful, so monitor flagging provides no safety benefit.
>
> **Would the results change or stay the same if both Blue team and Red team had access to the same number of prompts for validation?**
>
> We hope that our response above clarifies the purpose of the dataset splits. Especially: the Blue Team's use of the data serves a technical calibration purpose, rather than providing strategic advantage. We have updated the language in the paper to make this clearer.

---

> > ### Comment · Reviewer_XKux · 2025-08-04
> >
> > Thanks for your detailed response and clarifications. I will increase my score to accept.

---

### Official Review · Reviewer_WFZb · 2025-07-04

**Clarity:** 3
**Significance:** 2
**Originality:** 1
**Rating:** 5
**Confidence:** 4

**Summary:**

This paper evaluates automated monitoring of language models, both on the basis of their final outputs (the action-only monitor) and also on the basis of both the chain-of-thought (CoT) and its final outputs (the CoT monitor). They introduce a hybrid monitor, which essentially just averages the scores of the other two monitors. In each case, a threshold for monitor suspicion is selected during the validation phase. The authors evaluate all three monitors using the BigCodeBench instruct dataset, augmented with side-tasks and “attack policies” that give the tasked model some direction to help it evade the monitor.  They find a clear clustering between “subtle side-tasks” that are more easily detected by the CoT monitor, and “non-subtle” side tasks that are more easily detected by the action-only monitor. They find that the hybrid monitor outperforms both approaches on both types of tasks. These results hold across six different reasoning models and “attack policies”. All evaluations are on the same dataset and with the same monitor model (GPT 4.1-mini).

**Questions:**

I have a number of questions about your weights and averaging process:

* How do your results change across values of w? I was hoping for a table or figure in section A.4, but there’s nothing there.  Specifically, I would be interested in seeing the results for w = 0.25, 0.5, 0.75.

* Why do you think $w > 0.5$ is optimal given that this puts less weight on the monitor that has strictly more information?

* The false positive methodology is non-standard. Why not report F1 (or even classification accuracy) rather than going through all the effort described in appendix A.3? Have you tried setting the threshold to a fixed value and reporting simpler metrics?

* Based on your finding that CoT monitors are more effective on subtle tasks and action-only monitors are more effective on non-subtle ones, I expected the hybrid combination to be multiplicative rather than additive. Intuitively: if either monitor is sufficiently suspicious, then the hybrid score should be high.
More precisely, one could implement this with a (weighted) geometric average rather than a weighted arithmetic one:
$$\hat s_{\text{hybrid}} := 100 \left(1 - \sqrt{ \Big(1- \frac{s_{\text{action-only}}}{100} \Big)\Big(1-\frac{ s_{\text{CoT+action}}}{100} \Big) } \right).$$
Note that this formula is just
$s^\prime_{\text{hybrid}} = \sqrt{ s^\prime_{\text{CoT}} \cdot s^\prime_{\text{action}} }$
where $s' = (1-s/100)$ is the (normalized) score for innocuousness rather than suspiciousness, and has the property that both monitors have the power to to veto the verification. Have you thought about aggregation methods like this?

A few more questions:

*  Have you thought about replacing the GPT4.1 mini monitor model with small models from other families? I would imagine that CoT monitors from the same family would be more vulnerable to persuasion, and this might be worth studying.

* Can you provide some transcript instances where the agent successfully avoids detection in a subtle case? Based on the transcripts provided, it seems that these models explicitly recall their side objective in their chain of thought, so I’m very curious about why the CoT monitor sometimes fails to pick this up.


## Some Low-level Issues, Errors, and Comments

* line 126: based on the description in appendix A.3, there should not be a $\le$ ; the false positive rate should be exactly 1% on the calibration data, and so one imagines approximately 1% elsewhere. The inequality introduces some confusion about the methodology. This paragraph can be made clearer in other ways as well. It’s not just that you “set a threshold” (line 125) but rather that you set *the classification threshold*; at first read, this seems like another filtering step in the context of the previous paragraph, making it seem unnecessarily suspicious.

* In figure 6, the labels on the x-axis overlap, and are in the range [0,1] rather than [0,100] per the description of the monitor tasks A.1.

* The x-axis of figure 7 is mislabeled; its 1-FPR, not FPR.

* I reject the conclusion on lines 246-247: “as our results suggest, the adoption of 247 hybrid evaluation strategies that prevent over-reliance on a single information stream” — Your hybrid monitor is no different from the CoT+action monitor baseline in this regard; both have access to the same information. You just process it differently. For similar reasons, I think the wording on line 229 is misleading.

* In my view, the blue team / red team material is rather generic, not very interesting or technical, and does not add much to the paper. I find the typeset colors distracting, and create an unintended association between the red/blue in figures 3 and 5.

**Ethical Concerns:**

["NO or VERY MINOR ethics concerns only"]

**Final Justification:**

* The additional analysis provided by the authors in the rebuttal to myself and to reviewer VKc9 allayed biggest technical concerns about how the results generalize across multiple settings and about the fair comparison per unit of computation. I hope that ultimately the entire evaluation of the hybrid monitor will be done with this methodology.
* Keep in mind that the authors must now add a section to motivate their higher-temperature run-twice methodology and compare that itself against the baseline. I feel there is a significant opportunity to strengthen the result by adding further baselines in this spirit (e.g., using a model of twice as many parameters, or prompting the  model to think twice as long to double check).
* I still believe that the title is inappropriate and that the framing is wrong in some places---I have tried to explain why, but ultimately this is the authors choice.

Ultimately, reflection, discussion with the authors, and reading the other reviews, have convinced me that the authors are in a position to make this paper worth publishing by the camera-ready deadline.

**Limitations:**

The authors’ limitations section is good.

**Quality:**

3

**Strengths And Weaknesses:**

# Strengths:
The finding that side-tasks cluster nicely into subtle and non-subtle objectives depending on how easy they are to detect based on the final response alone, appears to be an important contribution of this paper. The appendix clearly explains and defends the methodology (and more of this material should be integrated into the main body of the paper.) The work finds similar findings across six families of frontier models. The extremely simple nature of the hybrid monitor makes its effectiveness surprising and raises interesting scientific questions. These surprising empirical findings and experimental hygiene are clearly the strength of the paper.

# Weaknesses:
The threshold and weight tuning methodology appears to be more convoluted than it needs to be, and there is some key information missing. The proposed hybrid monitor is, on its face, rather substance-less: it is just an average of two scores, based on the finding that sometimes the monitor with less information works better. The fact that this outperforms the model with more information is interesting, but, to my mind, this raises more fundamental questions about the setup more than it supports the hybrid-monitoring solution. The evaluation is run on only one dataset (albeit with many different tasks), and with one monitor model, raising questions about whether or not the seemingly robust behavior illustrated in the paper holds in related settings. Given the simplicity of the model and the strangeness of the behavior, a lot of weight falls on thorough evaluation across multiple settings—which fall slightly short of convincing, in my view.  I also really dislike the title.

A few specific concerns:

* While it makes sense that the monitors should be weaker models, I wonder if the results of figures 4 and 5 are simply a result of the model being too weak to competently understand the instructions and the thought process. I began to suspect this after reviewing the selected transcripts.

* Choosing $w = 0.5$ instead of $w=0.55$ would have significantly simplified the presentation and methodology, allowing the authors to remove appendix A.4 and the paragraph from lines 129-134. This suggests to me that the method is very sensitive to the choice of $w$. However, results are reported only for $w=0.55$, which I find suspicious.

* Since the hybrid monitor comprises both the CoT and action-only monitors, it uses roughly twice as much computation. Therefore, one could argue that there is a need for a fairer (compute-normalized) baseline—maybe using a 2x larger model for the individual monitors or averaging the scores for two distinct monitors of the same type.

* The title is too bold for the contributions of this paper. The chain-of-thought monitoring is not being stress-tested at all—indeed, the solution is to have a monitor that often ignores the chain of thought! The “CoT red-handed” supposed to be a pun involving chain-of-thought and the notion of a red-team, but the focus of this paper is on the “blue team” and shifting weight away from a potentially convincing CoT. In any case, the title does not accurately reflect the interesting empirical finding of the paper: that sometimes CoT monitoring is useful (for subtle tasks) and other times it is counter-productive.

---

> ### Author Rebuttal · Authors · 2025-07-31
>
> We thank the reviewer for their thoughtful comments. We appreciate the feedback and aim to address the mentioned weaknesses and questions that they bring up.
>
>
> ## Rationale for Hybrid Monitoring Strategy
>
>
> We chose to pursue the hybrid strategy of weighting the action-only and CoT monitor scores primarily due to our empirical observation that the action-only and CoT monitors had complementary strengths on different types of side tasks.
>
> The fact that the hybrid monitor outperforms the monitor with "all" information (i.e., both the action and CoT) was surprising to us at first as well, but we believe that the rationale is that the misleading nature, or overload of information of the CoT leads to underperformance of the CoT monitor in some cases. We do agree that the approach of simply weighting the action-only and CoT scores is quite naive, and we imagine that there are likely better approaches to aggregating monitor inputs; to this end we have investigated the reviewer’s suggested geometric weighting (see below).
>
> We hope that our exploration of this method in the control setting will be useful as a source of information to inform much more thorough design work in future monitoring systems.
>
>
> ## Single dataset and monitor model
>
>
> We accept that the strength of our paper would be improved with studies over more datasets and monitor models. To address this, we have run additional experiments with further monitor models: 4.1-nano, Claude 3.5 Haiku and Gemini 2.5 Flash Lite (resoning). On the subject of multiple datasets, we did explore several sets, including Mostly Basic Python Problems, APPS and CWeval, but in the end chose to focus on BigCodeBench as it was the most extensive and highest in quality. We think it fair to highlight that we study both a range of side tasks and a range of untrusted models. Further to this, many important works in AI control, such as the original AI control paper, only work with a single dataset and a single monitor model.
>
>
>
>
> ## Expanded Monitor Evaluation, weight values, suggested geometric average
>
> As per the reviewer's note, we believed that it was important to keep the monitor weaker than the scheming reasoning models. We agree that our results suggest that the monitor may simply have been too weak to do monitoring well, so in order to corroborate the results on GPT-4.1 mini, the monitoring model in our original paper, we have tested three other monitor models which were weaker than our scheming reasoning models: Claude 3.5 Haiku, Gemini 2.5 Flash Lite (reasoning), and GPT-4.1 nano.
>
> We ran monitoring with these models, as well as a second run with GPT-4.1 mini, and replicated our results finding that the setting of the weighting value to w=0.55 outperforms or performs on par with other weightings, including 0.5, as well as with other scoring mechanisms, i.e., using the action-only score, the CoT score, and the reviewer's suggested geometric mean.
>
>
> ## Performance Results Across Multiple Monitors
>
> **Catch Rate @ 1% FPR**
>
> | Monitor | Action-Only | CoT-Only | Hybrid (w=0.25) | Hybrid (w=0.50) | Hybrid (w=0.55)     | Hybrid (w=0.75) | Max(Action,CoT) | Geometric Mean |
> | --- | --- | --- | --- | --- | --- | --- | --- | --- |
> | claude-3.5-haiku | 45.95% | 30.76% | 40.04% | 45.03% | 46.5% | 49.45% | 44.12% | 45.23% |
> | gemini-2.5-flash-lite | 68.82% | 66.64% | 68.24% | 80.16% | 80.63% | 76.96% | 82.34% | 80.98% |
> | gpt-4.1-mini | 27.9% | 23.9% | 25.12% | 30.19% | 31.01% | 29.91% | 32.96% | 31.79% |
> | gpt-4.1-nano | 21.72% | 7.85% | 6.66% | 12.67% | 13.71% | 16.17% | 15.98% | 13.73% |
>
> **AUC-ROC Scores:**
>
> | Monitor | Action-Only | CoT-Only | Hybrid (w=0.25) | Hybrid (w=0.50) | Hybrid (w=0.55) | Hybrid (w=0.75) | Max(Action, CoT) | Geometric Mean |
> |---------|-------------|----------|-----------------|-----------------|-----------------|-----------------|------------------|----------------|
> | Claude 3.5 Haiku | 0.7688 | 0.7465 | 0.8142 | 0.8210 | 0.8211 | 0.8186 | 0.7830 | 0.8218 |
> | Gemini 2.5 Flash Lite | 0.8578 | 0.8596 | 0.8862 | 0.9037 | 0.9045 | 0.9019 | 0.9002 | 0.9035 |
> | GPT-4.1 Mini | 0.7395 | 0.7492 | 0.7758 | 0.7764 | 0.7765 | 0.7759 | 0.7758 | 0.7764 |
> | GPT-4.1 Nano | 0.6590 | 0.5933 | 0.6466 | 0.6476 | 0.6477 | 0.6476 | 0.6475 | 0.6476 |
>
> **F1 Scores at 1% FPR:**
>
> | Monitor | Action-Only | CoT-Only | Hybrid (w=0.25) | Hybrid (w=0.50) | Hybrid (w=0.55) | Hybrid (w=0.75) | Max(Action, CoT) | Geometric Mean |
> |---|---|---|---|---|---|---|---|---|
> | claude-3.5-haiku | 0.6276 | 0.3157 | 0.3918 | 0.5138 | 0.5476 | 0.5388 | 0.4876 | 0.5140 |
> | gemini-2.5-flash-lite | 0.5013 | 0.3658 | 0.5334 | 0.5641 | 0.5964 | 0.5873 | 0.4652 | 0.6267 |
> | gpt-4.1-mini | 0.4362 | 0.4959 | 0.5437 | 0.5562 | 0.5561 | 0.5278 | 0.5368 | 0.5562 |
> | gpt-4.1-nano | 0.3559 | 0.3353 | 0.3577 | 0.3668 | 0.3867 | 0.3716 | 0.3603 | 0.3668 |
>
> These results also reflect running the action-only and CoT monitors twice (with higher temperature of T=1.0, including our original GPT-4.1 mini runs), so that they have comparable resource usage to the hybrid monitoring method.
>
> 1. Model Capability Effects: The weaker GPT-4.1 nano monitor performs relatively worse compared to the other models, while our original GPT-4.1 mini monitor performs slightly worse than the more performant Claude 3.5 Haiku and Gemini 2.5 Flash Lite monitors.
>
> 2. Reasoning Model Performance: Gemini 2.5 Flash Lite is a reasoning model, meaning that it was trained to use its own CoT when generating outputs. We believe that this contributes to its relatively stronger performance compared to the other, non-reasoning models.
>
> 3. Alternative Scoring Methods: We note that the reviewer's suggestion of trying the geometric mean works well, performing on par with our tuned hybrid scoring, and we appreciate the contribution.
>
>
>
> ## Controlling for monitoring resources
>
> The reviewer is right that we should have controlled for monitoring resources by running two repeats of the action-only and CoT-and-action monitors for fair comparison to the hybrid monitor. We have addressed this experimentally - please refer to our above results and our response reviewer VKc9 for a description.
>
>
>
> ## Title criticism
>
>
> We respectfully disagree with the criticism of our title. The pun is meant light-heartedly, but we do include work on several red-team strategies, and our ‘stress-testing’ successfully finds places where CoT monitoring does not work well relative to action-only monitoring (alongside situations where it works better).
>
>
>
> ## Optimality of w=0.55
>
> We determined this empirically for our particular set of results - we wouldn't claim that future projects expanding on these ideas should adopt this particular number as generally applicable. We think that the reason for the heavier weighting on action monitoring is the same as the reason that we describe for CoT monitoring underperforming in some cases, i.e misleading content in the CoT.
>
>
>
> ## Phrasing on hybrid monitor performance
>
> We agree. Thank you for pointing this out - we’ve changed the wording on those lines to reflect the real advantage of the hybrid monitor, i.e that the different approaches perform better on different side tasks, and the hybrid monitor takes into account both. This is what we were trying to express, but we agree our original wording is imprecise.
>
>
> ## Blue team/red team material
>
> Thanks for this - we agree and have changed this section to make it more concise.
>
>
> ## Thresholding methodology
>
> Thank you for bringing this up. We have clarified this paragraph in our redraft, including your phrase ‘set the classification threshold’.
>
> The inequality is necessary because of a feature of the suspicion scores we observed our monitor to produce. While we asked for a score between 0 and 100, and hoped that the returned scores would be roughly continuous, we actually observed that the returned scores mostly fell into buckets at 0, 5, 10, 15, and so on.
>
> This is likely to be explained by the fact that humans would usually use these milestones to assign such scores in GPT-4.1’s pre-training set. We carried out some experiments to try to change this behaviour and achieve a more continuous score, including changing the range, and adding explicit instructions to the prompt, but were unable to change this behaviour.
>
> This being the case, the 99th percentile, calculated across suspicion scores of the validation set without a side task, was often precisely at one of these milestones; several examples might score exactly at this level rather than above or below it.
>
> To deal with this, we decided that the monitor should only flag with suspicion scores strictly greater than this threshold were considered suspicious when the monitor was used in anger; i.e examples with suspicion scores exactly at this threshold were not considered suspicious. This seemed like the cautious approach, and maps to a false positive rate that is less than or equal to 1%.
>
>
>
> ## Axis labels in Figure 6
>
> Thank you for highlighting this - we have corrected this.
>
> ## Other questions
> Request for additional transcripts: Transcripts are too long for rebuttal. We will update our transcripts website after the rebuttal period ends to avoid running afoul of NeurIPS rules on external content.
>
> # Summary of Changes
>
> 1. Language revision to temper claims about performance differences
> 2. Expanded monitor evaluation including multiple models and sensitivity analysis
> 3. Added methodological details about prompt development and optimization
> 4. Clarified limitations regarding OpenAI reasoning models
> 5. Explicit discussion of score combination strategies
> 6. Comprehensive performance comparison across multiple monitors and scoring methods
>
> We believe these changes significantly strengthen the paper while maintaining our core contributions about the relative effectiveness of different monitoring approaches, the vulnerability of simple CoT monitors to reasoning content, and the benefits of combined monitoring strategies.

---

> > ### Comment · Reviewer_WFZb · 2025-08-06
> >
> > Thank you for the thorough rebuttal. You have addressed all of my major concerns, and will be increasing my score. Upon reflection, I am coming to appreciate the finding, and I can increasingly see its practical value.  The new experiments have shored up my concerns about fair baselines.
> >
> > Nevertheless, I still think its packaging could be improved.  The finding that the hybrid approach works is presented in the paper as a solution to the problem that CoT+action and action-only monitors both fail.  But the method itself is not so interesting---what's worth stressing, to my mind, is the higher-level scientific finding that **ignorance of the chain of thought  model can have a benefit (in non-subtle tasks), and that this benefit can be combined with the benefits of knowing the chain of thought**.  The way in which the combination happens is kind of irrelevant, as evidenced by the many variants of the experiments in the tables of the author's rebuttal to my review.
> >
> > I attribute my somewhat-negative initial reaction to the fact that the paper seems to try to "sell" the hybrid-monitor---which is simple-minded and un-principled, amounting to averaging information X to information which should intuitively be a subset of X---as a robust and principled solution. This just raises questions about why this should work and whether something sketchy is going on beneath the surface.  I feel all of these concerns would go away if the authors were to more fully emphasize their surprise that this works at all, and leave open the question about why we see this effect and how universal it is.
> >
> > This ties in with my concern about the title: "CoT Red Handed" sells a monitor (that uses CoTs, presumably the hybrid monitor), which is ok, but what I think should be the emphasis of the work. On its own, the second half of the title ("stress-testing chain-of-thought monitoring"), more accurately reflects what I see as the value in the paper: the detailed analysis of the two kinds of monitors, and the surprising observation that their strengths can be combined in a simple way.  However, when the first half of the title is in scope, the meaning of the second half changes---it now suggests to me that the authors have stress-tested the new CoT-based monitor that they are selling and have verified that it works in a wide array of extreme circumstances. This interpretation is what I find disingenuous. Yes, you have stress-tested CoT monitors, but the outcome was not pretty; meanwhile, the reason you can claim to "catch scheming red-handed" is because of a different model (the hybrid monitor) that  works by leveraging **ignorance** of the CoT.
> >
> > As a result, I feel the title contaminated my interpretation of the value of the entire paper.  I hope you can fix this while keeping your light-hearted pun---but it's not worth torquing the entire paper to fit the joke in the title.

---

> ### Author Response · Authors · 2025-08-06
>
> Thank you again for your thoughtful comments on our submission. We noticed you haven't yet responded to our rebuttal, and we'd appreciate knowing whether it addressed your concerns or if there is any additional information we could provide.
>
> We have also provided detailed rebuttals to all other reviewers and those may answer any lingering questions.

---

> ### Comment · Area_Chair_J2aX · 2025-08-06
> **Please reply to authors**
>
> Dear reviewer, please reply to the authors' rebuttal.
>
> I'd also like to invite you to replace your wording when talking about a "rather stupid" approach (in the weakness section of your review). You may change stupid to straightforward not to offend anyone and keep a professional tone. Thanks.

---

### Official Review · Reviewer_VKc9 · 2025-07-05

**Clarity:** 2
**Significance:** 3
**Originality:** 3
**Rating:** 5
**Confidence:** 5

**Summary:**

The paper evaluates whether CoT monitoring improves monitor performance, compared to action-only monitoring, at detecting malicious side-task attempts on a coding benchmark. They find that CoT helps the monitor overall, but can also be used to mislead it. Therefore, the paper also proposes a hybrid monitoring approach combining CoT and action only monitoring and demonstrates its efficacy.

**Questions:**

- Do you agree that the comparison for hybrid monitoring is unfair for the reasons I discuss? Can you address this concern?
- Are the side tasks an original contribution?
- Can you deal with my structure concerns?
- Can you execute any of the additional experiments I suggested?

**Ethical Concerns:**

["NO or VERY MINOR ethics concerns only"]

**Final Justification:**

Overall the authors have compellingly addressed my concerns and I will increase my score to an accept. I have also read the other reviews in detail (see my comments below). I agreed with several points raised by other reviewers but I think the authors have addressed all the major concerns raised. I do not feel there are any other substantial weaknesses of the paper.

**Limitations:**

yes

**Quality:**

3

**Strengths And Weaknesses:**

**I will increase my score if my two major concerns (see below) are addressed.**

Strengths:

- Well motivated direction and clear scope of experiments
- well written paper
- comprehensive empirical evaluation on many models and tasks
- comprehensive and mostly clear methodological details


Weaknesses

Major

- Poorly structured (e.g., essentially three methods sections) and the content is not really well-ordered, making some details hard to find or unclear. (See below)
- Unfortunately I think there is a substantial methodological flaw in one of the core results, BUT I think it can hopefully be addressed.  Unless I’m mistaken, the hybrid monitor results do not control for monitor resources, so the comparison to action only and CoT monitoring is unfair. (I guess a similar point could be made for CoT vs action only.) Perhaps a really principled way to deal with this would be to control for the monitor input token budget or something like that. However, a simpler way to get a fairer comparison would be to just allow action and CoT monitoring to also average over two predictions (with some non deterministic monitor). What I mean is that, any monitor will do better if you sample several scores and average them (cf jury theorems). And the hybrid monitor basically gets to sample two judges, whereas the others just get one sample score. I’d be happy to increase my score if you account for this concern (e.g., by rerunning the action-only and CoT monitor results with high temperature and getting results for weighted average over, e..g, 2 action-only monitor scores).

Other comments.

Figure 1 is nice

The related work doesnt seem that informative. E..g, you discuss alignment faking in too much detail, and miss some control references like ShadeArena. The lightman citation also seems a bit tangential. I think the related work could be clearer and more organised. Perhaps it should also go later in the paper. In general, the rel work section can be used to either contrast your work to other work (then it should go near the end) or to set the scene and provide greater context (then it should go near the beginning).

Sections should never begin with a subsection (e.g., Background) they should begin with a short overview of the section.

I think some of the content can be moved around, e.g., the CoT part of the background should possibly go in related work (at least the first paragraph). The background is kind of a weird short section. I would remove it and incorporate the content into the related work and methods. Also, the discussion of the dataset should come much sooner, like the first thing in methods. The experimental setup section should be folded into methods too (with subsections). Id basically start methods with discussion of the dataset and side tasks then discuss control and red team blue team roles and protocols.

It’s unclear in some places what’s novel and what isnt. For instance, did you come up with the side tasks or are they from other work? Also, how are the side tasks graded? With test cases?

In methods you should discuss how you evaluate performance, both of the agent (at completing main and side tasks) and especially of the monitor — Fig 3 reports the catch rate, I would have gone with ROC-AUCs which is pretty standard in control work right?

I understood that you remove unsuccessful agent attempts and only monitor successes, but it could be interesting to see how the results look for unsuccessful sabotage attempts.

Figure captions should be entirely self-contained (yours are too minimal).

The figures seem a bit out of place, e.g, figure 2 is not discussed until near the end of the paper (I would discuss it sooner).

I would be very interested to see results comparing CoT monitor performance on reasoning models (CoT optimised with RL) vs non-reasoning models. Cf https://arxiv.org/abs/2501.08156

Summary of extra experiment suggestions:

- fair comparison between hybrid protocol and the others (by sampling multiple monitors for the others)
- see if the results hold for sabotage attempts not just successes
- compare CoT monitoring for reasoning vs non-reasoning models to see whether optimised CoT influences monitorability

Please also report ROC-AUCs, at least in the appendix, which I think are more informative than the 1% FPR catch rate (which seems like a somewhat arbitrary choice?)

---

> ### Author Rebuttal · Authors · 2025-07-31
>
> We thank the reviewer for the thoughtful and detailed feedback. We appreciate the constructive suggestions and recognize the importance of addressing the methodological and structural concerns raised.
>
> ## Methodological Fairness of Hybrid Monitor Comparison
>
> We agree with the reviewer that in our original comparison the hybrid monitor benefited from multiple samples while the baseline monitors only had access to one.
>
> To address this concern, we ran both action-only and CoT monitors twice with temperature set to 1.0 and averaged their suspicion scores, while the hybrid monitor used only a single run for each component. All results included in this response or the responses to other reviewers use this methodology.
>
> We acknowledge that the action-only monitor processes fewer tokens than the CoT monitor, but we believe this is a fundamental trade-off rather than a disadvantage. The action-only monitor focuses on the most relevant information (i.e. the final implementation), and an ideal monitor should be capable of detecting sabotage attempts from this output alone.
>
> Additionally, following reviewer suggestions, we evaluated three additional monitor models, including both reasoning and non-reasoning models (detailed results are provided in other rebuttals):
>
> | Monitor Model | Action Catch Rate | CoT Catch Rate | Hybrid Catch Rate |
> |-|-|-|-|
> | claude-3.5-haiku | 45.95% | 30.76% | 46.50% |
> | gemini-2.5-flash-lite | 68.82% | 66.64% | 80.63% |
> | gpt-4.1-mini | 27.90% | 23.90% | 31.01% |
> | gpt-4.1-nano | 21.72% | 7.85% | 13.71% |
>
> These results largely confirm our original findings: the hybrid approach consistently outperforms both baseline strategies across most monitor models. However, for the weakest monitor (GPT-4.1-nano), the hybrid approach shows limited improvement, suggesting that a minimum level of monitor capability may be required for the hybrid strategy to be effective.
>
> We note that due to resource constraints, we used the same w=0.55 value (optimized for the original run of GPT-4.1-mini on our validation set) across all monitor models rather than optimizing individually. This suggests the hybrid approach's benefits could potentially be even stronger with model-specific tuning.
>
> **Related Work Organization and Content**
>
> We appreciate the feedback on our related work section and understand the concern about its organization and focus. Our aim for this section is to provide context about existing applications of CoT monitoring and demonstrate how our work differentiates from them.
>
> Our inclusion of alignment faking work reflects its direct relevance to CoT monitoring in adversarial settings, particularly given that Greenblatt et al. specifically examine how models can use reasoning traces to deceive monitors. We believe this reference establishes important context for understanding the challenges our work addresses.
>
> Regarding the Lightman et al. citation, we included this work because it represents a foundational approach to supervising reasoning processes in language models, albeit for training rather than deployment monitoring.
>
> We acknowledge that ShadeArena was published after our submission and therefore could not be included in our original related work. We would be happy to incorporate discussion of this and other relevant concurrent work in a revised version.
>
> ## Paper Structure and Clarity
>
> We understand the reviewer's concerns about the paper's structure and will work to address these issues. Our current organization reflects our view that the background section provides essential context about CoT and the AI control framework that readers need before understanding our methodological contributions.
>
> In our experimental setup section, we explain the particular instantiation of the method we are studying in our experiments. We describe the dataset in the experimental setup because it represents a specific feature of our experiments rather than a core aspect of our methodology. Although we make an effort to ensure that the side tasks we define in the method section are as relevant as possible to the samples of our selected dataset, they could be used with multiple coding and cybersecurity benchmarks.
>
> However, we will revise the paper structure to improve clarity and ensure that each section has an introductory overview.
>
> ## Novelty and contributions
>
> As outlined in our introduction, we explain our main contributions to distinguish our novel work from existing research. Regarding the side tasks, these represent a combination of our novel contributions and adaptations from existing work.
>
> Most control work (e.g. Greenblatt et al.) has focused on logical errors as proxies for backdoors, which corresponds most closely to our "edge case" side task that creates behavioral differences with the gold standard solution. Several other tasks draw inspiration from existing literature: the exit side task is inspired by reward hacking strategies that models exploited in Baker et al., while the download side task is similar to scenarios examined in Bhatt et al. Additional side tasks, such as exfiltrate secret, are motivated by discussions in the AI Safety community regarding primary threat models.
>
> Our primary contribution in developing these side tasks lies in collecting and implementing them within a single unified environment. This allows us to study to characterize when different monitoring strategies are most effective.
>
> As mentioned in Appendix A.2, side tasks are evaluated using test cases. For each side task, we developed specific judging criteria tailored to the task's characteristics. For example, the complexity side task is evaluated by measuring and comparing the running times of both the gold standard solution and the red team solution, the variable name task uses regex matching to detect the specified variable name, and the download file task verifies both the presence and content of the downloaded file. We will provide comprehensive details of the evaluation method of each side task in the revised paper.
>
> ## Choice of Evaluation Metrics
>
> The reviewer suggests using ROC-AUC instead of catch rates at 1% false positive rate (FPR). While we appreciate this suggestion, we respectfully maintain that our chosen metric better reflects realistic deployment constraints. ROC-AUC averages performance across all possible FPRs, including unrealistically high rates that would never be acceptable in practice. In real-world deployment scenarios, organizations would likely operate at very low FPRs due to the substantial costs of human review and intervention.
>
> Our 1% FPR choice reflects the practical reality that monitoring systems must operate under strict precision constraints when dealing with large volumes of queries and limited human oversight resources. This threshold represents a reasonable balance between detection capability and operational feasibility, making our results more directly applicable to real deployment scenarios.
>
> Nevertheless, for completeness and to address the reviewer's suggestion, we provide ROC-AUC results across different monitor models. These results confirm the improvement of the hybrid monitor (w=0.55) over either baseline metric except for the weakest monitor (GPT-4.1-nano). However, the difference between performance according to AUC is much closer than with catch rate at 1%FPR, indicating that the model might not underperform as much at lower FPRs.
>
> | Monitor Model | Action AUC | CoT AUC | Hybrid AUC |
> |-|-|-|-|
> | claude-3.5-haiku | 0.7688 | 0.7465 | 0.8211 |
> | gemini-2.5-flash-lite | 0.8578 | 0.8596 | 0.9045 |
> | gpt-4.1-mini | 0.7395 | 0.7492 | 0.7764 |
> | gpt-4.1-nano | 0.6590 | 0.5933 | 0.6477 |
>
> As for the 1%FPR metrics, we used the same w=0.55 value across all monitor models rather than optimizing individually.
>
> ## Filtering by Task Success
>
> Our decision to filter results to include only successful completions of both main and side tasks serves important methodological purposes. For main task failures, including these would introduce noise that obscures our core research question about monitoring effectiveness, as monitors might flag outputs due to general incompetence rather than malicious intent.
>
> For side task filtering, side task failures can occur for two distinct reasons: genuine attempts that fail due to implementation errors, or complete non-compliance where models refuse to attempt the side task due to safety training. The latter case creates an unfair comparison between action-only and CoT monitoring, as action-only monitors cannot reasonably be expected to detect sabotage attempts that exist only in reasoning traces without any corresponding implementation.
>
> While we recognize that including failed attempts could provide additional insights, we believe our current approach ensures a fair comparison between monitoring strategies by focusing on cases where malicious behavior is actually present in the final output.
>
> ## Reasoning vs Non-Reasoning Models
>
> We appreciate the suggestion to compare monitor performance across reasoning and non-reasoning models. Our focus on reasoning models reflects their position at the current frontier of capabilities where our monitoring approaches would be most relevant.
>
> Non-reasoning models were considered early in our research but ultimately excluded from our final evaluation. Based on existing research showing reduced faithfulness in prompted CoT compared to trained reasoning, we expect that non-reasoning models would be more susceptible to certain evasion strategies, such as simple instructions to avoid mentioning side tasks in scratchpad reasoning. This would fundamentally change the threat model and reduce the relevance of our findings to future deployment scenarios where reasoning models are likely to predominate.
>
> We remain open to exploring this comparison in future work, as it might provide insights into CoT faithfulness and monitoring effectiveness across different architectures.

---

> > ### Comment · Reviewer_VKc9 · 2025-08-02
> > ****Overall the authors have compellingly addressed my concerns and I will increase my score to an accept. I have also read the other reviews in detail (see my comments below). I agreed with several points raised by other reviewers but I think the authors have addressed all the major concerns raised. I do not feel there are any other substantial weaknesses of the paper.****
> >
> > **Overall the authors have compellingly addressed my concerns and I will increase my score to an accept. I have also read the other reviews in detail (see my comments below). I agreed with several points raised by other reviewers but I think the authors have addressed all the major concerns raised. I do not feel there are any other substantial weaknesses of the paper.**
> >
> > > Methodological Fairness of Hybrid Monitor Comparison
> >
> > The authors straightforwardly and compellingly addressed my concern on this point by running the additional experiments I requested.
> >
> > > Choice of Evaluation Metrics
> >
> > The authors convinced me that the catch rate @ 1% FPR is the correct metric for this setting. I think the ROC-AUCs should be included in the appendix.
> >
> > > Filtering by Task Success
> >
> > Okay seems fair. Possibly I still think the former case (not refusals) is good to show.
> >
> > > Reasoning vs Non-Reasoning Models
> >
> > I think you should note this as an area of future work — it might be especially interesting exactly because non reasoning models are less faithful (if thats the case)
> >
> > > Paper Structure and Clarity
> >
> > I think your paper will be more impactful if you do make efforts to substantially improve these points :)
> >
> >
> > Below I leave some comments on my agreements and disagreements with the other reviewers.
> >
> >
> > > Reviewer WFZb
> >
> > - I think the reviewer raises good points regarding the hybrid monitor weighting methodology (but I think the authors sufficiently addressed this in their rebuttal to the reviewer). I also liked the reviewers suggestion for a geometric weighting (which the authors demonstrated works)
> > - This reviewer shared my concern regarding monitor resources, but I feel the authors have completely addressed this in their rebuttal to me
> > - I am not concerned about limited evaluation on different benchmarks. I think the range of experiments conducted on bigcodebench is a sufficient contribution (of course, one can always improve by including more benchmarks, but I don’t feel this is a major limitation of this paper.)
> > - I like the title and think it fairly represents the paper
> > - “The false positive methodology is non-standard.” I think it is fairly standard in the AI control literature, and I also now agree with the authors that it makes sense as a metric, in any case the authors now included ROC AUCs too
> >
> >
> > On the weighting: I guess here is one thing you would want to do instead: because the action and CoT monitor perform better or worse on subtle vs non subtle tasks, you would want to train a model to classify subtlety vs non-subtley (really: classify which monitor performs better), then you weight according to this classifiers probability over subtlety. This might be an interesting idea for future work :)
> >
> > Also, did you optimise the weight on a “train set” of tasks? Probably you care more about this, since we want monitors to generalise to unseen tasks.
> >
> > > Reviewer XKux
> >
> > I think the reviewer raises a good point about some of the subtle side-tasks not appearing very malicious or suspicious. However, I think this is a bit difficult to get right. Which actions are suspicious can be highly context-dependent (as the authors point out, some of the non-obviously harmful side tasks may form part of a broader misaligned objective).
> >
> > > Reviewer 9pfw
> >
> > The reviewer was concerned that they only evaluate one monitor model, but the authors now validated their results on many monitor models in the rebutal.
> >
> > I didnt really agree that the authors overstated the performance gap between action and CoT monitors (relative performance is still different even iof absolute for both is low). Agree that absolute performance is too low for reak life control and the authors could note this. Agree that prompt optimisation is an important consideration — I think its a bit difficult to do right, but putting the same amount of effort into each method, and utilsiign a train set for prompt optimisation can help.

---

### Decision · Program_Chairs · 2025-09-17

**Decision:**

Accept (poster)

**Comment:**

The paper proposes CoT monitoring to detect misbehavior or sabotage of AI models. There is a unanimous consensus towards acceptance.